# Current Hepatitis C Vaccine Candidates Based on the Induction of Neutralizing Antibodies

**DOI:** 10.3390/v15051151

**Published:** 2023-05-11

**Authors:** Elsa Gomez-Escobar, Philippe Roingeard, Elodie Beaumont

**Affiliations:** Inserm U1259 MAVIVH, Université de Tours and CHRU de Tours, 37000 Tours, France; elsa.guadalupe.gomez.escobar@umontreal.ca

**Keywords:** hepatitis C virus, vaccine development, neutralizing antibodies, envelope glycoproteins

## Abstract

The introduction of direct-acting antivirals (DAAs) has revolutionized hepatitis C treatment. Short courses of treatment with these drugs are highly beneficial to patients, eliminating hepatitis C virus (HCV) without adverse effects. However, this outstanding success is tempered by the continuing difficulty of eradicating the virus worldwide. Thus, access to an effective vaccine against HCV is strongly needed to reduce the burden of the disease and contribute to the elimination of viral hepatitis. The recent failure of a T-cell vaccine based on the use of viral vectors expressing the HCV non-structural protein sequences to prevent chronic hepatitis C in drug users has pointed out that the induction of neutralizing antibodies (NAbs) will be essential in future vaccine candidates. To induce NAbs, vaccines must contain the main target of this type of antibody, the HCV envelope glycoproteins (E1 and E2). In this review, we summarize the structural regions in E1 and E2 proteins that are targeted by NAbs and how these proteins are presented in the vaccine candidates currently under development.

## 1. Introduction

The World Health Organization (WHO) estimated that 58 million people live with chronic hepatitis C and 1.5 million new infections with hepatitis C virus (HCV) occur annually [1,2]. Recent studies have shown that HCV elimination by 2030 might not be achieved unless measures for screening and treatment but also the prevention of HCV infection evolve globally [1,3,4,5]. Eight genotypes of this enveloped, positive-sense single-stranded RNA virus have been identified, but genotypes 1, 3, and 4 are alone responsible for 85% of the infections worldwide [6,7,8,9,10]. HCV infection can have two outcomes: spontaneous resolution or persistent infection. In around 75% of the cases, hepatitis C may become chronic leading to the development of severe liver diseases, such as hepatocellular carcinoma (HCC) [11,12]. In contrast, spontaneous resolution of HCV infection has been reported to occur in 25% of patients [13,14] and to be associated with the detection of early cellular and humoral responses [15,16]. This natural mechanism of infection control is an important hint in the development of a hepatitis C vaccine.

Current treatment based on direct-acting antivirals (DAAs) leads to the elimination of HCV in more than 95% of the cases, but it has some limitations. The risk to develop HCC after treatment remains high, especially in patients with advanced liver fibrosis [1,17,18,19,20,21]. Resistance to DAAs was also observed in subjects infected with some rare HCV subtypes (1l, 4r, 3b, 3g, 6u, 6v) that emerged in specific geographical zones [17,22,23]. DAAs do not protect from reinfection, and so intravenous drug users (IVDUs), which are frequently exposed to HCV, remain at high risk of reinfection after a DAA-mediated cure [24,25,26]. Beyond that, DAA-based treatments remain expensive for patients if not covered by a national program or by personal medical insurance. A recent study in the USA showed that treatment rates varied considerably by age and insurance payor with the result that only one-third of patients with an HCV diagnosis and medical insurance receive DAA treatment [27]. Thus, the generation of a vaccine against this virus will help to control its transmission, such as in high-risk populations, and respond to the current limitations of treatment [28,29,30].

To date, the most advanced HCV vaccine candidate is a T-cell vaccine consisting of a prime-boost regimen with two different viral vectors that encode the genotype (Gt) 1b (BK strain) HCV non-structural proteins NS3-5B (mutated in the NS5B gene to abolish the RNA polymerase activity) (NSmut) [31]. This vaccine was first shown to induce strong cellular responses in chimpanzees and protect 80% of them from challenges with HCV [32]. Then, the safety and efficacy of various viral vectors encoding the NSmut construct (adenovirus 6, chimpanzee adenovirus 3 (ChAd3), and modified vaccinia Ankara (MVA)) were evaluated in healthy volunteers (clinical trials: NCT01070407 and NCT01296451) [33,34]. Barnes and collaborators found that the vaccine was well tolerated with no severe adverse effects and led to the generation of cellular responses, especially when using the MVA-NSmut, as a booster, which induced strong and sustained CD4+ T cell responses over time [33,34]. However, in the latest randomized clinical trial phase 1/2 (NCT01436357), in which 274 participants (IVDUs) followed the prime-boost regimen ChAd3-NSmut/MVA-NSmut, vaccination did not prevent the development of chronic infection [35]. These results suggest that humoral responses characterized by broadly neutralizing antibodies (bNAbs), along with cytotoxic and helper T cell responses, as well as the conception of novel immunogens that generate immune responses against genetically diverse HCV genotypes/subtypes should be considered in HCV vaccine development, as discussed in a recent review [36]. Our review focuses on the structural components needed for the induction of neutralizing antibodies (NAbs) and the current status of HCV envelope-based vaccine candidates aiming to elicit humoral responses.

## 2. The Envelope Glycoproteins as the Target of Neutralizing Antibodies

The HCV envelope glycoproteins (E1 and E2) constitute the main targets of NAbs. These proteins are highly glycosylated (5-6 and 11 N-glycans, respectively) transmembrane proteins type I, anchored to the endoplasmic reticulum (ER)-derived membrane by a 30-amino acid (aa) transmembrane domain (TMD) and located at the surface of HCV [37]. These proteins can interact with each other and form large covalent complexes stabilized through disulfide bonds on the surface of virions, or non-covalent heterodimers intracellularly [38]. Several groups have characterized the structure of the HCV envelope proteins individually, but lately, the structural characterization of the full-length E1E2 heterodimer was achieved and it proposes novel structural elements to consider for the HCV vaccine development. In this review, E1 and E2 proteins are numbered according to the HCV polyprotein (isolate H77; GenBank: AF009606) [39] unless indicated otherwise.

### 2.1. The Envelope Glycoprotein E1

In the HCV viral cycle, the E1 protein participates in the viral entry by interaction with cellular receptors, such as claudin (CLDN)-1 [40,41], CLDN-6 [42,43], and CD36 [44]. It has also been suggested that E1 contributes to the fusion step due to a region identified as a putative fusion peptide (FP) (Figure 1), regardless of the structural differences with fusion proteins characterized by other viruses of the *Flaviviridae* family [45]. The crystal structure of the full-length E1 protein was unknown until not long ago because the expression of E1 in the absence of the E2 protein can lead to protein aggregation [46,47,48]. Thus, El Omari and colleagues solved the N-terminal domain of a Gt 1 (H77 strain) E1 protein through crystallization in low-pH conditions, representing a post-attachment conformation of the domain (Figure 1) [49]. This study reported that dimers of the crystallized protein did not resemble the classical fusion proteins from flaviviruses, instead, they were rather small and structurally similar to the phosphatidylcholine transfer protein [49]. So, it was proposed that this structure may bind the apolipoproteins E (apoE) and B [50]. However, in the latest study by Torrents de la Peña and collaborators [51], the conformation of the N-terminal domain of E1 protein, within the heterodimer, differed from the structure determined by El Omari and colleagues [49]. This finding may confirm that E1 requires E2 protein for correct folding. The study also confirmed the presence of 4 disulfide bonds and the potential N-glycosylation sites with the exception of N325 (Figure 1), previously described to be absent when a proline residue is present immediately following the sequon (Asn-X-Ser/Thr) [52]. The E1 protein has as well the ability to form trimeric structures through the GxxxG motif located within its TMD [53], the same motif that was suggested to be involved in the interaction between the TMDs of both envelope proteins [54,55]. This trimeric arrangement of the E1E2 heterodimers has been confirmed by using computational and biological models [56,57] as well as suggested by the lack of glycans in a hydrophobic region on the structure of the recently solved E1E2 complex [51].

The envelope protein E1 is one of the targets of NAbs, but only a few of these antibodies are directed to this protein compared to the great number of E2-derived NAbs that have been characterized. Two main immunogenic regions in the E1 protein inducing NAbs have been identified: the N-terminus between aa 192–207 [60,61] and the conserved region between aa 313–328, near the C-terminus [58,62]. The human monoclonal antibody (mAb) H-111 (linear epitope aa 192–202) (Figure 1) was isolated from a Gt 1b HCV-infected subject presenting high titers of antibodies directed against the envelope glycoprotein E1. However, this antibody showed weak neutralizing activity when used to neutralize HCV pseudoparticles (HCVpp) [61]. Regarding the region between aa 313–328, the mAbs IGH505 and IGH526 were identified to target this site and shown to neutralize Gt 1a and 2a HCVpp, as well as HCV generated in cell culture (HCVcc) [62]. The structure of this epitope (aa 314–324) complexed with the mAb IGH526 was solved by X-ray crystallography and found to be discontinuous within mostly E1 of the E1E2 heterodimer (Figure 1) [58]. While the epitope of the mAb IGH505 was defined in complex with the E1E2 heterodimer and found to target the surface-exposed conserved α-helix in E1 (H316, W320, M323, M324) [51]. Because of the location of the epitopes of both antibodies (IGH526 and IGH505) in E1, it was proposed that neutralization may occur by impeding conformational changes in the heterodimer [51]. Aside from these two regions, Colbert and colleagues identified the antigenic site (AS) 112, spanning aa 215–299, targeted by the conformation-dependent NAb HEPC112 (Figure 1), which was able to neutralize 7 strains of HCV Gt 1a using the HCVpp system [63]. The mAb A6, characterized by Mesalam and colleagues, also targets a linear epitope within the AS112 (aa 230–239). However, this antibody isolated from Gt 1b HCV-infected patient does not exhibit neutralizing activity [64].

### 2.2. The Envelope Glycoprotein E2

The E2 protein participates actively in the entry step of the viral life cycle by interacting with the scavenger receptor-class B type I (SR-BI) [65,66] and the CD81 receptor [67]. The interaction between E2 and the CD81 receptor was recently characterized by Kumar and colleagues who proposed a docking model in which the residues 418–422 in E2 are displaced and allow the extension of an internal loop spanning the residues 520–539, which approaches Tyr529 and Tyr531 to the membrane in preparation for a low-pH-mediated fusion [68]. In a complementary study, Kumar and colleagues reported that for proper interaction with CD81, the front layer and the AS412 in E2 are essential [69]. The structure of the E2 ectodomain alone from different HCV genotypes was solved in complex with various antibodies [70,71,72,73,74]. Kong and collaborators solved the structure of a Gt 1a E2 core (aa 412 to 645) in a complex to the fragment antigen binding (Fab) of the bNAb AR3C (Figure 2) [70]. They observed a well-defined globular structure of the E2 core with a central Immunoglobulin (Ig)-like β-sandwich (aa 492–566) flanked by a front (aa 424–459) and a back layer (aa 597–645), and some disordered regions. They also identified most of the glycans of the E2 protein, except for N417, N448, N476, and N576, suggesting that these missing glycans may mask neutralization epitopes [70]. Khan and collaborators also solved the structure of a Gt 2a E2 core protein, but in complex with the Fab of the non-neutralizing 2A12 antibody (Figure 2). They observed the same globular structure as well as some glycans in the E2 core structure: N540, N556, N623, and N645 [71]. Additional studies reported the co-crystallization of the E2 ectodomain with other bNAbs and suggested significant flexibility in the structure of this protein [72,73,74]. The structure of the full-length E2 protein was recently determined within the E1E2 heterodimer Gt 1a by Torrents de la Peña and collaborators [51]. This study resolved the structure of 2 new regions in the E2 protein: the base (aa 645–700) and the stem (aa 701–717), which connects the base with TMD. It also confirmed that E2 has 9 disulfide bonds, consistent with previous studies [72], but differed from the structure obtained in a complex by Kong and collaborators due to disulfide-bond scrambling [75]. As reported for E1, they observed as well all the potential N-glycosylation sites in E2. However, a noncanonical NXV motif at the N695 glycosylation site was reported with and without binding to AR4A and AT1209 Fabs, and its removal led to a slight increase in viral infectivity using HCVpp [51].

The E2 protein is the target of most of the NAbs described for HCV, which may be explained by its accessibility on the surface of virions, in contrast to the E1 protein that may be shielded by E2 and apolipoproteins [57,76]. The E2 protein is composed of 3 variable regions, which span around 30% of the protein and help HCV to escape from the immune responses [77]. The hypervariable region 1 (HVR1) is a 27-aa long sequence located at the N-terminus of the E2 protein (Figure 2) that exhibits a high degree of variability among genotypes and subtypes. This immunodominant region can elicit NAbs and non-NAbs [78,79,80]. The binding of weak- or non-NAbs to the HVR1 could lead to the blocking of conserved neighbor sites, such as the epitope I [81]. Interestingly, the deletion of the HVR1 was reported to increase the susceptibility to neutralization by NAbs of virions, suggesting a shielding effect of antigenic sites in the E2 protein by the HVR1 [82,83]. In a recent study of a vaccine candidate based on inactivated recombinant HCVcc, deletion of the HVR1 led to increased accessibility of NAbs (AR3A and AR4A), but did not result in increased immunogenicity suggesting a much more complex role of this region [84,85].

**Figure 2 viruses-15-01151-f002:**
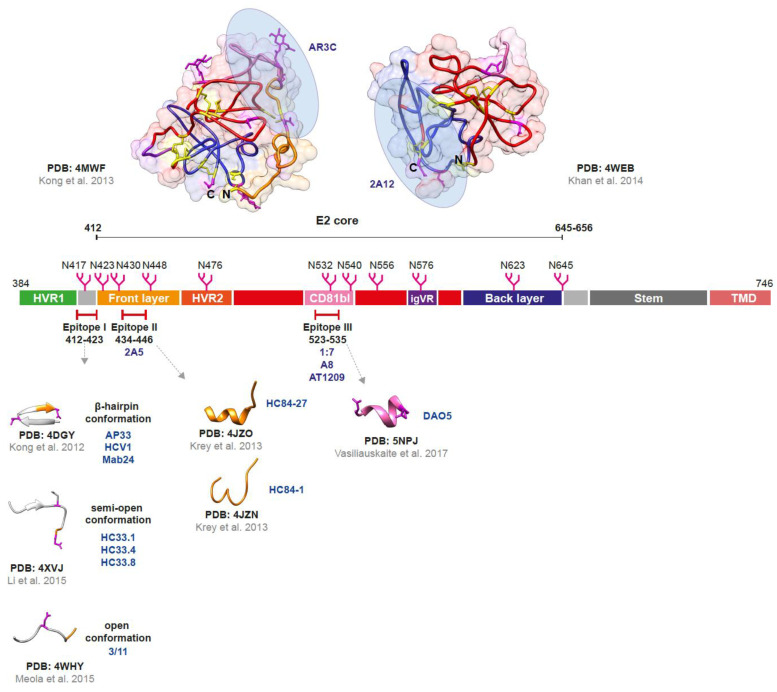
Structure of the HCV envelope glycoprotein E2. E2 is a 360 aa protein (including 30 aa for the transmembrane domain (TMD)) that contains 3 variable regions (hypervariable region (HVR)1, HVR2, and intergenotypic variable region (igVR)), a front layer, a back layer, a CD81-binding loop (CD81bl), a stem region and 11 N-glycosylation sites (in magenta). The crystal structure of the core of E2 considered individually (aa 421–645/656) was obtained by two groups (PDB: 4MWF and 4WEB). Three epitopes for neutralizing antibodies were identified (I, II, and III). The epitope I (aa 412–423) can adopt 3 conformations: β-hairpin (PDB: 4DGY), semi-open (PDB: 4XVJ), and open (PDB: 4WHY). The epitope II (aa 434–446) was co-crystallized with the human monoclonal antibodies (mAbs) HC84-27 (PDB: 4JZO) and HC84-1 (PDB: 4JZN), but also targeted by the 2A5 mAb. The crystal structure of the epitope III (aa 523–535), inside the CD81bl, (PDB: 5NPJ) was also obtained by co-crystallization with the mouse mAb DAO5, and targeted by the mAbs 1:7 and A8. Antibodies targeting each epitope are indicated in dark blue. Cysteines (and disulfide bonds) are represented in yellow and asparagines in magenta [70,71,75,86,87,88,89]. Molecular graphics were performed using Chimera UCSF [59]. PDB: Protein Data Bank (https://www.rcsb.org/ accessed on 26 February 2023).

Three epitopes named I, II, and III, have also been identified as targets of NAbs isolated from HCV-infected subjects. The epitope I (aa 412–423), also known as AS412, is a conserved region found downstream of the HVR1 containing 2 N-glycosylation sites (N417 and N423) (Figure 2). Antibodies directed to this site are elicited in only 2.5 to 15% of the patients [90,91]. This epitope is characterized by its flexibility, and so 3 different conformations have been described: the β-hairpin, the open, and the semi-open conformations (Figure 2) [86]. The β-hairpin conformation is stabilized through hydrogen bonds and side chains with hydrophobic interactions [75]. MAbs targeting this conformation are well characterized: the mouse mAb AP33 (aa 412–423) [75,92,93,94], the human mAb HCV1 (aa 412–423) [95,96], and the mouse Mab24 (aa 411–428) (Figure 2) [97]. The HCV1 and AP33 mAbs are broadly neutralizers that were shown to protect from the challenge after passive immunization of chimpanzees [98] or mice with a humanized liver [99], respectively. However, the escape from neutralization has been reported for each of them (HCV1, AP33, and Mab24) due to a glycan shifting (N417 to N415) in the antigenic site [100,101]. The semi-open conformation was described through the crystallization of the epitope I complexed with the human mAb HC33.1 (aa 412–423), an antibody isolated from a donor with a chronic HCV infection (Gt 2b isolate) [102]. This antibody was shown to neutralize HCV despite mutations and glycan shifts that prevent neutralization with mAbs AP33 and HCV1 [87,103]. However, the binding of other NAbs from the same panel (H33.4 and H33.8) was inhibited by the binding of antibodies to the neighbor region HVR1 [81]. The open conformation was characterized thanks to the rat mAb 3/11 whose binding to the epitope was shown to be sensitive to denaturation [86,94,104]. This conformation was observed on the surface of virions during the entry, but was not as predominant as the β-hairpin, contributing to the evasion of humoral responses [86].

The epitope II (aa 434–446) is composed of a 1.5 α-helix turn followed by an extended segment and flanked by the glycosylation sites N430 and N448 (Figure 2). The glycosylation site N448 was reported to reduce sensitivity to neutralization and to be involved in the reduction of the infectivity of HCV in vitro [105]. Different results were reported regarding the efficacy of combinations of antibodies targeting either epitope I or epitope II. Tarr and collaborators reported that when combining anti-epitope I and anti-epitope II antibodies, a better in vitro neutralization of HCV was achieved [90], while Zhang and colleagues found that non-NAbs binding the epitope II could prevent the neutralization of NAbs targeting epitope I [106]. To evaluate the escape to neutralization of antibodies targeting the epitope II, Keck, and colleagues used a set of 9 human mAbs, designated HC84, which can neutralize HCVcc and HCVpp. The culture of HCVcc with these antibodies resulted in no viral escape, while the culture with antibodies targeting the epitope I led to viral escape [107]. Krey and collaborators used the human mAbs HC84 to crystalize the epitope II of a Gt 1a E2 protein (Figure 2) and found that residues 435, 436, 439, and 443 were highly conserved, which may be determinant for the binding and the broad neutralizing activity of these antibodies [88]. Another antibody targeting this site is the mAb 2A5 (conformational epitope within the aa 434–446 region) which was isolated from a patient chronically infected with Gt 1b HCV. This antibody neutralizes several genotypes and strains of HCV in the HCVpp and HCVcc systems, as well as protects humanized mice against HCV challenge [108].

The epitope III (aa 523–535) is located inside the CD81 binding loop (CD81bl) (Figure 2). Because of the importance of this site in the viral entry step through the CD81 receptor, blocking the E2-CD81 interaction is a main target of bNAbs. In the study from Torrents de la Peña and collaborators, the bNAb AT1209, which targets a region comprising the CD81bl and the front layer, was used to determine the structure of E1E2 [109]. They revealed that the aa critical for the interaction between E2 and CD81 overlap with those located in the epitope of AT1209 [51]. Among the bNAbs targeting the CD81 are those reported by Johansson and colleagues, mAbs 1:7 and A8, which were isolated from a Gt 2b HCV-infected subject and able to recognize an epitope in the region spanning residues aa 523–535 [110]. This conserved region of the E2 protein has been shown to be highly flexible, as it can be recognized by antibodies approaching the protein at different angles [111]. Vasiliauskaite and collaborators designed a peptide from the Ig-like β-sandwich of the CD81bl that, in complex with a mouse anti-E2 mAb (DAO5) (Figure 2), adopted an α-helix conformation, suggesting that the CD81bl may fold in at least two conformations. The authors also showed that the CD81bl of E2 proteins on the surface of HCVpp may be recognized by mAbs binding the different conformations of this region [89]. Moreover, it has been proposed that the binding to this CD81bl site may require the recognition of aa in distant regions in E2, such as the front and back layers [112,113]. For instance, mutations in residues in the back layer alter the folding of the E1E2 heterodimer and thus, its reactivity to human mAbs with neutralizing properties [112].

Antigenic regions (AR1-5) [114,115] and domains (A-E) have been also described in the structure of the E2 protein [61,102,107,116,117,118]. These two nomenclatures overlap in residues with the previously described epitopes, but domains and regions are mainly defined on a conformational basis (Table 1). For instance, the antigenic domains B and D overlap with the AR3, while the antigenic domain E corresponds to the epitope I (or AS412) [103]. The AR4 contains as well epitopes of bNAbs like AR4A, which was used in the characterization of the full-length E1E2 heterodimer. Torrents de la Peña and colleagues reported that the E1E2 complex bound to AR4A was stabilized in a prefusion conformation suggesting that neutralization with this antibody occurs by hindering conformational changes needed for fusion [51]. The domain C overlaps with the AR5, but antibodies targeting the domain C do not bind to the E1E2 heterodimer unlike those targeting the AR5 [61]. The AR3 is a highly conserved region that induces the generation of bNAbs able to block the E2-CD81 binding [115]. This region, comprising the CD81bl and the front layer, is also called the neutralizing face of E2 because of the broadly-neutralizing nature of antibodies induced [119]. However, a study showed that a rare mutation in E1 may induce resistance to the bNab AR3A [120]. Thus, the design of immunogens presenting these regions, domains, and epitopes of bNAbs should be considered for the development of vaccines aiming to elicit humoral responses.

### 2.3. The E1E2 Heterodimer

The structural characterization of the E1E2 heterodimer has been hindered by the challenging purification process of these transmembrane proteins. So, the structure of truncated versions of E1E2 and in silico models was proposed. That is the case of Cao and collaborators that designed a functional heterodimer with E1E2 ectodomains (Gt 1b, ﻿strain Con1) fused with an Fc-tag; this complex could be recognized by NAbs such as AR3A, HCV-1, and IGH526, and bound some cellular receptors [122]. The in silico models predicted the E1E2 complex structure using computational tools such as Rosetta [56,57]. Other studies aimed to determine the role of every residue in the HCV envelope glycoproteins following mutagenesis strategies [112,123]. For instance, Gopal and colleagues found out that residues in the back layer of E2, within the heterodimer, can impact the binding to CD81 and NAbs [112]. A similar comprehensive mutagenesis study reported that 57% (311/545) of the mutants resulted in non-functional HCV envelope glycoproteins and 92% of the mutations led to a change in viral functionality, suggesting that the high rate of replication of HCV must be a mechanism to compensate the vulnerability of its envelope glycoproteins [123]. 

Lately, Torrents de la Peña and collaborators determined the structure of the full-length E1E2 heterodimer Gt 1a (AMS0232 strain) in complex with Fabs from AR4A, AT1209, and IGH505 NAbs (Figure 3) [51]. In this study, the authors were able to model 51% of E1 and 82% of E2 comprising the E1/E2 interface, the disulfide networks of E1 and E2, the E1/E2 glycan shield and the epitopes of the NAbs used, while the unresolved TMDs were predicted using AlphaFold (Figure 3A). Torrents de la Peña and colleagues proposed a “stem-in-hand” model in which the ectodomain of the E1 protein wraps the E2 stem and interacts with the base of E2 (Figure 3B). They reported that the non-covalent interactions (hydrophobic and hydrogen bonds) observed between E1 and E2 were consistent with previous studies [124,125] and that the glycans N196 and N305 reinforced this interaction. Despite the prior identification of highly conserved cysteine residues, the disulfide patterns were different from previously determined structures. Thus, the authors hypothesized that the proximity of cysteines in the HCV envelope glycoproteins may allow disulfide bond scrambling [51]. These findings will contribute to the design of immunogens possessing the best structural characteristics to induce bNAbs.

## 3. Analysis of Antibodies from HCV-Infected Subjects

In HCV infection, antibodies have been reported to appear from week 5 to 8 post-infection [126,127]. Spontaneous clearance of HCV infection has been associated with the early appearance of antibodies while patients with chronic infections have been shown to develop antibodies at later stages of the disease [14,15,16,128,129]. However, the precise role of antibodies in the course of HCV infection remains unclear. NAbs isolated from subjects who spontaneously cleared the infection and from chronic carriers have been characterized, and found to both target the same epitopes on the neutralizing face of the E2 protein [72,109,114,115,130,131]. Some of these antibodies have been defined as broad neutralizers and reported to protect from challenges with homologous and heterologous HCV strains [98,114,115,132] or even abolish the infection [133].

Plasma from a large cohort of patients in the early stages of HCV infection was analyzed and bNAbs isolated were preferentially directed towards the AR2, AR3, AR4, and domain D, suggesting that these sites should be considered for the design of immunogens for an HCV B-cell vaccine [134]. Another study reported that from a cohort of chronically HCV-infected subjects, less than 5% developed bNAbs whose neutralizing properties were linked to the use of the heavy chain variable region (V_H_) 1-69 segment, to somatic mutations within the complementarity-determining region of the heavy chain (CDRH) 1 and CDRH2 hydrophobicity [135]. Furthermore, a comparative study characterizing the immune response induced by a prototype vaccine (recombinant E1E2 heterodimer from Dr. Michael Houghton’s team) reported that most of the antibodies elicited by the vaccine were weak- or non-NAbs directed against regions such as the HVR1 [136]. However, AR3-derived antibodies were capable to cross-neutralize HCVcc and HCVpp from different genotypes, and encoded by a set of genes from V_H_ with 90% homology to the V_H_1-69 lineage in humans [136]. Interestingly, bNAbs directed against the neutralizing face of E2 have been previously reported to be mainly derived from the same lineage (V_H_1-69) [72,74,130,135]. The V_H_1-69 set of genes, characterized for presenting a few somatic hypermutations and suggesting a rapid lineage development, was also identified for antibodies involved in the clearance of other viruses such as HIV [137,138] and influenza viruses [139,140]. Therefore, it may be important that antibodies elicited by HCV vaccine candidates share those characteristics (lineage and AR3-directed). 

Despite the generation of antibodies with broad neutralizing properties during HCV infection, escape mechanisms from humoral responses have been reported. Bailey and collaborators found that the substitutions I538V, Q546L, and T563V in the E2 protein (from a library of 19 Gt 1 HCV sequences) conferred resistance to neutralization by modification of the folding of the protein, even if the binding residues were found outside the epitopes of NAbs [141]. These results are supported by other studies demonstrating that distant mutations can lead in particular to conformational changes of epitope I and CD81bl, affecting the binding of antibodies and inducing resistance to NAbs [82,86,89,142,143]. The high level of glycosylation of the E2 protein is another documented escape mechanism due to the shielding of epitopes that prevents the binding of antibodies [105,144]. However, Khera and collaborators showed that mutation in the glycosylation sites and the HVR1 in a recombinant E2 vaccine induced cross-binding antibodies, but without cross-neutralization properties, suggesting that something more than exposed epitopes is needed to elicit bNAbs [145]. The glycan shifting, a specific mechanism of escape, occurs due to a change in the glycosylation from N417 to N415 due to the mutations N417S or N417T in the E2 protein [101].

HCV circulates in the bloodstream of infected patients as a lipoviral particle because it can associate with host-derived lipids. This association was found to be another mechanism to evade the humoral response [146,147]. Bankwitz and collaborators found that apoE enhances the infectivity of viral particles and reduces the potency of NAbs to inhibit viral entry without occluding epitopes in E1 and E2 proteins [146]. These findings were in line with those reported by Fauvelle and colleagues who observed that the mutation F447A modifies the conformation of the E2-apoE complex, and thus modulates the sensitivity to NAbs [147]. Thus, vaccine development based on envelope glycoproteins should not only consider the structural complexity of E1 and E2 proteins but also the possible escape mechanisms (mutations, glycans, and apolipoproteins association) from humoral responses [148].

## 4. B-Cell Vaccine Candidates against HCV 

The characterization of the HCV envelope glycoproteins has facilitated the design of vaccines that elicit humoral responses. Here, we present an updated summary of vaccine candidates classified according to the immunogen selected to induce NAbs (Table 2): (i) E1 and E2 proteins together, (ii) E2 protein alone, or (iii) peptides/epitopes of the HCV structural proteins.

### 4.1. Vaccines Using Both HCV Envelope Glycoproteins (E1 and E2)

One of the pioneer B-cell vaccine candidates for HCV and the only one that has been evaluated in a clinical trial is the recombinant E1E2 heterodimer, developed by the team of Dr. Michael Houghton, awarded the Nobel Prize in Physiology or Medicine 2020. The vaccine is based on Gt 1a (HCV-1 strain) E1 and E2 proteins purified from mammalian cell cultures (Chinese hamster ovary (CHO) cells). Immunization of chimpanzees with this vaccine induced strong humoral responses that protected them from challenges with HCV, and when low antibody titers were elicited, the resolution of acute infection was merely delayed [149]. The antibodies induced by immunization were also reported to cross-neutralize HCV in vitro (HCVpp and HCVcc) [150]. In the clinical trial phase 1 (NCT00500747), immunization of healthy subjects with the E1E2 heterodimer, adjuvanted with MF59, resulted in the generation of antibodies and proliferation of T helper cells [151]. However, antibodies from half of the samples were reactive against the HVR1 and only around 50% of participants developed antibodies able to cross-neutralize HCVcc harboring HCV envelope proteins from various genotypes (genotypes 1a, 1b, 2a, 4a, 5a, and 6a were better neutralized than genotypes 2b, 3a and 7a) [152,153,154]. Immunization of mice with the vaccine candidate deleted from its HVR1 did not induce antibodies with better cross-neutralizing properties in comparison to the wild-type (WT) E1E2 proteins [155], but in HCVcc the HVR1 was found to be a major determinant in the sensitivity to neutralization with sera from immunized mice [156].

Because the envelope glycoproteins E1E2 are retained in the ER membrane, their purification remains a challenge, and several research groups have proposed the fusion of these proteins into molecules to improve their purification by maintaining their conformation. For instance, the E1E2 heterodimer was expressed with a Flag tag, which was reported to facilitate the purification step without affecting the heterodimerization and folding of the proteins or the CD81 binding site [157]. Immunization of C57BL/6 mice with the E1E2-Flag led to the generation of antibodies that were able to neutralize Gt 1a and 2a HCVcc with around 60% efficacy [157]. Lin and colleagues evaluated the fusion of an E1E2 heterodimer (from Gt 1a, 1b, 2a, 3a, and 6a codon-optimized sequences) to a human IgG1 Fc fragment, and found that the conformation of proteins was not altered [158]. Immunization of BALB/c mice with the adjuvanted pentavalent vaccine (Fc-E1E2 heterodimers of 5 genotypes) elicited NAbs with better neutralizing capacities against HCVpp (genotypes 1a, 1b, 2a, 2b, 3a, 4c and 5a) and HCVcc (Gt 2a) in vitro systems than the monovalent vaccines [158]. Another example is the case of a native-like soluble E1E2 glycoprotein, in which the TMDs of proteins were replaced by a leucine zipper scaffold and a furin cleavage site between E1 and E2 [159]. This vaccine was compared to the recombinant membrane-bound E1E2 heterodimer designed by Houghton and colleagues. The soluble E1E2 heterodimer was recognized by bNAbs targeting the antigenic domains B, D, and E at similar levels as the membrane-bound E1E2 vaccine candidate. After immunization of CD-1 mice, the soluble E1E2 heterodimer-induced antibodies targeting the E1 protein (H-111 epitope), and the domain B, D, E, AR4, and AR5 in the E2 protein, were analyzed through competitive inhibition assays. Neutralization of HCVpp and HCVcc was reported to be at similar or higher levels than those elicited by the membrane-bound E1E2 vaccine [159,160]. 

Another platform frequently used for the presentation of immunogens is the virus-like particles (VLP), in which proteins self-assemble to generate empty structures physically resembling virions. Because of their geometry, epitopes displayed across the surface of VLPs have been shown to strongly stimulate B cell responses [161]. Garrone and collaborators reported the production, in human embryonic kidney (HEK293T) cells, of retroviral VLPs pseudotyped with E1 and/or E2 proteins [162]. The immunization of macaques with these particles elicited antibodies that cross-neutralize HCVpp (genotypes 1a, 1b, 2a, 2b, and 4c), but anti-E1 antibodies were difficult to achieve [162]. Another strategy was based on HCV VLPs resembling HCV virions and containing structural proteins (core, E1, and E2) from a Gt 1a HCV produced in the human hepatoma (Huh7) cell line [163,164,165]. In BALB/c mice, these particles induced specific CD8+ T cells and antibodies able to neutralize Gt 1a HCVpp [165]. Moreover, vaccination of mice and Landrace pigs with HCV VLPs carrying structural proteins from 4 different genotypes (1a, 1b, 2a, and 3a) led to HCV-specific cellular and humoral responses, and the induced antibodies were able to neutralize Gt 2a and 3a HCVcc [163,164].

Following a similar strategy, our research group developed an original vaccine candidate based on the hepatitis B virus (HBV) small envelope protein (S), which can self-assemble into VLPs. We used the full-length Gt 1a HCV envelope glycoproteins (E1 and E2) fused to the heterologous HBV S protein to generate chimeric HBV-HCV proteins (E1-S and E2-S). These proteins co-expressed with the WT S protein in mammalian cells (CHO cells) can self-assemble into secreted and highly immunogenic subviral particles (S + E1-S or S + E2-S SVPs) [166]. We reported that immunization of New Zealand rabbits with chimeric HBV-HCV SVPs induced humoral responses against HBV and HCV, and that the antibodies were able to neutralize Gt 1a, 1b, 2a, 3a and 4a HCVpp and HCVcc [167,168]. We also showed that the use of a cocktail of particles bearing chimeric E2-S proteins of different genotypes (1, 3, and 4) elicited antibodies with significantly improved neutralization properties against Gt 3a and 4a HCVcc, compared to the group immunized with Gt 1a S + E2-S particles [169]. We also generated HBV-HCV vaccines bearing the apoE to mimic the interactions at the interface of apoE-HCV envelope glycoproteins as observed on the surface of virions. We showed that the detection of the chimeric E1-S and E2-S proteins by well-characterized antibodies was altered in the presence of apoE and that after immunization with the apoE-bearing particles, specifically, antibodies induced with S + E2-S + apoE SVPs showed better neutralizing potential against Gt 1a and 2a HCVcc [148].

The nanoparticles are commonly used as scaffolds for the display and delivery of immunogens to antigen-presenting cells (reviewed in [170]). In their study, Sliepen and collaborators proposed trimeric permutated E2E1 Gt 1a (strain AMS0232) nanoparticles [171]. HCV envelope glycoproteins were permutated to bring closer the N-terminus of E1 and the C-terminus of E2 in order to facilitate interactions between E1 and E2 proteins [51]. E2E1 nanoparticles were used to immunize New Zealand female rabbits, which developed antibodies able to neutralize more HCVpp than a monomeric E2 protein. E2E1 mosaic nanoparticles were also efficiently generated (harboring E2E1 from different genotypes: strains H77, AMS2b, AMS3a, UKNP4.1.1, UKNP5.2.1, UKNP6.1.2) with the objective of focusing the responses to epitopes that are conserved among the antigens presented. After immunization, the mosaic nanoparticles induced slightly better cross-neutralizing antibodies (6 HCVpp) than a cocktail of 6 E2E1 nanoparticles harboring each of the proteins from one strain [171]. 

The usage of inactivated pathogens constitutes another traditional approach to the development of vaccines [172]. In the case of HCV, inactivated HCV vaccine candidates have been proposed thanks to the HCVcc system [173]. The UV-inactivated vaccine developed by Akazawa and collaborators was based on a chimeric Gt 2a replication-deficient HCVcc [174,175]. These particles were produced in large-scale Huh7.5.1 cell cultures and purified over a sucrose cushion in ultracentrifugation with a recovery rate of 15% [176]. Immunization of BALB/c mice and nonhuman primates (marmosets *Callithrix jacchus*) with inactivated HCVcc induced cellular responses and antibodies against HCV structural proteins that cross-neutralize HCVpp (genotypes 1a, 1b, and 2a) and HCVcc (genotypes 1a, 1b, 2a and 3a) [175,176]. One advantage reported with this vaccine over a recombinant protein is that lower doses of inactivated HCVcc were required for the induction of antibodies [175]. Antibodies induced by immunization of human liver chimeric urokinase-type plasminogen activator-severe combined immunodeficiency (uPa-SCID) mice protected them from HCV challenge with the low viral doses (10^3^ RNA copies) [175]. However, these HCVcc displayed heterogeneous densities because of the association with lipids and apolipoproteins present in the animal-derived serum used in mammalian cell cultures. 

In line with this, another inactivated vaccine candidate was recently reported. HCVcc was produced under serum-free conditions in Huh7.5 cells and purified as established in a previous study [177]. Immunization with UV-inactivated HCVcc prepared with AddaVax^TM^ adjuvant (analog to MF59) induced strong bNAbs in comparison to the one adjuvanted with Alum + Monophosphoryl lipid A (MPLA) [85]. Genotypes 1a, 2a, and 3a HCVcc were produced in high-yield cultures by serial passages of HCV-engineered recombinants which led to an increase in NAbs epitopes exposure [84]. Complete neutralization of HCV in vitro (genotypes 1–6) was obtained with 1000 µg/mL purified IgG from mice immunized with UV-inactivated engineered Gt 1a, 2a, and 3a HCVcc [84].

### 4.2. Vaccines Using the HCV Envelope Glycoprotein E2

Because most of the epitopes of NAbs in HCV have been identified in the E2 protein, several vaccine candidates only contain this protein as an immunogen. Nevertheless, the genetic diversity of HCV remains an obstacle to the development of an effective vaccine. Thus, the use of proteins from different HCV genotypes in vaccination has become a common approach to overcome this challenge. 

An interesting vaccination strategy is the Gt 1b TMD-truncated soluble E2 protein produced in *Drosophila* S2 cells [178]. The glycosylation patterns generated by S2 cells on the soluble E2 protein are less complex than those generated in mammalian cells, which was suggested to increase the immunogenicity of E2 by improving the flexibility of the structure. Immunization of immunocompetent mice with these vaccine induced antibodies able to cross-neutralize genotype 1–7 HCVcc [178]. Because of the unclear effect of the HVR1 on the induction of NAbs, Li, and collaborators deleted this region on the soluble E2 protein (sE2), but no difference was observed in terms of immunogenicity compared to the one containing the HVR1 [178]. Antibodies induced by this vaccine candidate were characterized and resulted to be AP33-like and AR3A-like bNAbs. This vaccine was also used in combination with Alhydrogel 2% (aluminum-based) and MPLA adjuvants in the immunization of nonhuman primates (Rhesus macaques), leading to the induction of memory-type and interferon-γ-producing T cell responses, and NAbs [179]. Furthermore, the sE2 protein was evaluated as a cocktail of proteins from 3 genotypes (1b, 1a, and 3a), which elicited a better cross-neutralizing response for some HCVcc in comparison to the monovalent vaccine [180]. To improve the presentation of the E2 protein, it was coupled to ferritin self-assembling into nanoparticles, similar to VLPs. The conformation of sE2 was improved within the ferritin platform compared to sE2 alone, which enhanced its immunogenicity and the cross-neutralizing potential of elicited antibodies [181].

Another vaccine candidate based on E2 protein alone is the monomeric Gt 1a E2 protein deleted from the 3 variable regions (Δ123). Immunization of guinea pigs with the E2-Δ123 generated broad neutralizing responses (Gt 1–7 HCVcc) suggesting that deletion of the variable regions may have allowed the exposition of epitopes occluded in the native structure [182]. A multimeric form of this vaccine candidate was generated by sequential reduction and oxidation leading to the formation of disulfide bonds [183]. The production yield of the multimeric forms was improved compared to the monomeric ones, but well-described bNAbs showed reduced reactivity to the E2 protein in the multimeric form. Antibodies induced by the monomeric vaccine were reported to bind specifically the linear epitopes I, II, and III, and both forms of the protein elicited antibodies able to compete for CD81 binding similarly [183]. Moreover, the neutralizing potential of antibodies elicited with these particles by immunization of guinea pigs was only enhanced for Gt 1a HCVpp [183]. 

Likewise, rationally designed Gt 1a and 6a E2 core constructs, without regions inducing non-NAbs, were produced in mammalian cell cultures and associated with nanoparticles. Immunization of mice with nanoparticles containing Gt 1a or 6a E2 core elicited NAbs directed to epitopes in the front layer and the epitope I (or AS412), while the mix of Gt 1a and 6a nanoparticles led to low antibody titers and not better neutralizing potential than the single-genotype immunization strategy [184]. Tarr and colleagues developed another rationally-designed vaccine candidate based on the consensus sequence of the E2 core (ΔHVR1 and ΔC-terminus), from 720 strains of Gt 1 HCV produced in *Drosophila* S2 cells [185]. However, after immunization of guinea pigs, the elicited antibodies recognized mainly linear epitopes and neutralized exclusively Gt 1a HCVpp.

### 4.3. Vaccines Using Peptides or Epitopes of the HCV Proteins

In other cases, vaccine candidates against HCV have focused only on aa sequences or epitopes needed for the induction of NAbs. That is the case of the vaccine based on the HBV surface antigen (HBsAg) that can self-assemble into VLPs, and in which the hydrophilic loops were substituted by the epitopes I, II, or III of a Gt 1a HCV E2 protein. These chimeric particles were produced in *Leishmania tarentolae*, an expression system with a glycosylation pattern similar to mammalian cells [186,187]. Antibodies targeting epitope I (aa 412–423), purified from immunized mouse sera, showed better cross-neutralizing properties (Gt 1a, 1b, 2a, 2b, 4a, and 5a HCVcc) in comparison to antibodies targeting epitopes II and III [187]. Similarly, Wei and colleagues inserted epitopes from Gt 1a, 1b, and 2a HCV E2 proteins (including the HVR1) in the external hydrophilic loop of the HBsAg [188]. These particles were used individually or as a cocktail to immunize BALB/c mice. The highest titer of antibodies and the best cross-neutralization (Gt 1a, 1b, and 2a HCVcc and HCVpp) were obtained by the immunization with the cocktail of VLPs bearing different epitopes.

Dawood and collaborators selected peptides from Gt 4a HCV E1, E2, NS4B, NS5A, and NS5B proteins containing highly conserved residues among genotypes, and associated with the induction of strong B and T cell responses in spontaneous clearance [189]. Immunization of BALB/c mice with these peptides generated cellular and humoral responses (neutralization of Gt 2a and 4a HCVcc) in a dose-dependent manner. Similarly, two peptides, with high sequence divergence, from the HVR1 region of HCV isolated from patients were evaluated in mice. Vaccination with a mixture of both peptides resulted in better immune responses than each peptide individually and led to the cross-neutralization of HCVpp (genotypes 1a, 1b, and 6a were better neutralized than genotypes 2a, 3a, and 5a) [190]. 

**Table 2 viruses-15-01151-t002:** Current B cell vaccine candidates. HCV vaccine candidates are classified according to the immunogen chosen to elicit neutralizing antibodies: the E1E2 proteins, the E2 protein alone, and epitopes or peptides from HCV structural or non-structural proteins.

Immunogen	Vaccine Candidate	Genotype	Platform	Expression System	Models Tested	Antibody Characterization	Neutralization	Cellular Responses	References
E1E2	Recombinant E1E2/MF59 adjuvant	1a	Recombinant membrane-bound proteins	CHO cells	Chimpanzees Healthy humans	ELISA (anti-E1E2) and competitive immunoassays (epitopes: HVR1, AR3, AR4, domains C and D)	Gt 1-7 HCVpp and HCVcc	Yes	[149,150,151,152,153,154]
	E1E2-flag/IFA adjuvant	1a, 1b, 2a	Recombinant soluble proteins	HEK293T cells	C57BL/6 mice	ELISA (anti-E1E2)	Gt 1a and 2a HCVcc	NA	[157]
	Fc-E1E2/alum adjuvant	1a, 1b, 2a, 3a and 6a	Recombinant membrane-bound proteins	HEK293F cells	BALB/c mice	ELISA (anti-E1E2)	Gt 1a, 1b, 2a, 2b, 3a, 4c and 5a HCVcc	Yes	[158]
	Soluble native-like E1E2	1a	Recombinant soluble proteins	Expi293F cells	CD-1 mice	Competition inhibition analysis (epitopes: domains B, D and E, AR4, AR5 and E1)	Gt 1a HCVpp and HCVcc	NA	[159,160]
	MLV VLP E1E2	1a	VLP	HEK293T cells	BALB/c and C57BL/6J mice Cynomolgus macaques (*Macaca fascicularis*)	ELISA (anti-E1 and anti-E2)	Gt 1a, 1b, 2a, 2b and 4c HCVpp	Yes	[162]
	HCV VLP core, E1 and E2/alum adjuvant	1a, 1b, 2a, 3a	VLP	Huh7 cells	BALB/c mice Landrace pigs	ELISA (HCV VLPs)	Gt 1a, 2a and 3a HCVcc	Yes	[163,164,165]
	Bivalent chimeric HBV/HCV vaccine/AddaVax^TM^ adjuvant	1a, 3a, 4a	VLP	CHO cells	New Zealand female rabbits	ELISA (anti-E1 and anti-E2)	Gt 1a, 1b, 2a, 3a and 4a HCVpp and HCVcc	NA	[148,167,168,169,191]
	UV-inactivated HCVcc vaccine/K3-SPG adjuvant	2a	Inactivated HCVcc	Huh7.5.1 cells	Chimeric liver uPA^+/+^-SCID mice; Marmoseth (*Callithrix jacchus*)	ELISA (anti-core, -E1, -E2)	Gt 1a, 1b, 2a and 3a HCVcc	Yes	[175,176]
	Inactivated whole HCV vaccine/AddaVax^TM^ adjuvant	1a, 2a, 3a and 5a	Inactivated HCVcc	Huh7.5 cells	BALB/c mice	ELISA (anti-E2 and anti-E1E2)	Gt 1a, 1b, 2a, 2b, 3a, 4a, 5a and 6a HCVcc	NA	[84,85]
	E2E1-nanoparticle	1, 2, 3, 4, 5 and 6 (cocktail and mosaic)	Recombinant soluble proteins in nanoparticles	Suspension 293F cells	New Zealand female rabbits	ELISA (anti-E2 or anti-E2E1) and competitive ELISA (bNAbs)	Gt 1-6 HCVpp	NA	[171]
E2	Soluble E2/Ferritin/Alhydrogel+MPLA adjuvants	1a, 1b, 3a	Recombinant soluble proteins in nanoparticles	*Drosophila* S2 cells	BALB/c mice Rhesus macaques	ELISA (anti-E2) and competitive ELISA (AP33-like and AR3A- like bNAbs)	Gt 1-7 HCVcc	Yes	[178,179,180,181]
	E2 Δ123 variable regions/AddaVax^TM^ adjuvant	1a	Recombinant soluble proteins	FS293F cells	Albino Dunkin Hartley guinea pigs	Direct ELISA (anti-E2 and anti-E2 Δ123), capture ELISA (epitopes I, II and III) and competitive ELISA (CD81 binding)	Gt 1-7 HCVcc	NA	[182,183]
	Consensus core E2 ΔHVR1 ΔC-terminus	1a (720 strains)	Recombinant soluble proteins	*Drosophila* S2 cells	Guinea pigs	ELISA (anti-E2)	Gt 1a HCVpp	NA	[185]
	E2 core nanoparticles /AddaVax^TM^ adjuvant	1a, 6a	Recombinant soluble proteins in nanoparticles	HEK 293F and ExpiCHO cells	BALB/c mice	ELISA (anti-E2, and epitopes in front layer and AS412)	Gt 1a, 2a, 5a and 6a HCVpp	NA	[184]
Epitopes or peptides	HBV VLPs carrying HCV E2 protein epitopes/AddaVax^TM^ adjuvant	1a	VLP	*Leishmania tarentolae*	BALB/c mice	ELISA (anti-E2 epitopes 412–425 and 523–535)	Gt 1a, 1b, 2a, 2b, 4a and 5a HCVcc	NA	[186,187]
	Chimeric HBV S antigen VLPs presenting HCV-neutralizing epitopes/AddaVax^TM^ adjuvant	1a, 1b, 2a	VLP	HEK293T cells	BALB/c mice	ELISA (anti-HCV-neutralizing epitopes)	Gt 1a, 1b and 2a HCVcc and HCVpp	NA	[188]
	Multi-epitope peptide vaccine (E1, E2, NS4B, NS5A and NS5B)	4a	Synthetic peptides	Synthesis by the 9-fluorenylmethoxy carbonylmethod	BALB/c mice	ELISA (anti-HCV peptides)	Gt 2a and 4a HCVcc	Yes	[189]
	Bivalent HCV peptide (HVR1) vaccine/ Freunds complete or incomplete adjuvant	1a	Synthetic peptides	Synthesis using Fmoc chemistry	BALB/c mice	Competitive ELISA (HVR1, C-terminus)	Gt 1a, 1b, 2a, 3a, 4a, 5a and 6a HCVpp	NA	[190]

NA: not analyzed.

## 5. Conclusions

The failure of the viral-vectored T-cell vaccine to protect from progression to chronic hepatitis C suggests that NAbs directed against HCV envelope glycoproteins may be essential for a successful HCV vaccine. Thus, hepatitis C vaccine candidates should, from now on, be designed to induce a combination of broad humoral and cellular responses. A better understanding of the role of T and B cell responses during HCV infection will guide the design of future HCV vaccines. Herein, we presented a wide diversity of B-cell vaccine candidates based on the HCV envelope glycoproteins that were developed in recent years. However, it remains difficult to compare the efficacy of the antibodies induced by vaccination due to the lack of standardized in vitro assays. The establishment of HCVpp and HCVcc panels considering the resistance to neutralization of some strains may help to compare vaccination efficiency and, ultimately, to facilitate the pre-clinical validation [192,193]. Furthermore, findings from structural studies may contribute to improving the definition of NAbs epitopes induced by vaccination, which are frequently merely characterized as anti-E1/anti-E2 (Table 2). Finally, the recent publication of the E1E2 heterodimer structure will enable the design of better immunogens containing the structural elements needed for E1E2 complex stabilization and epitopes presentation [51], such as the E2E1 nanoparticles-based vaccine candidate that was designed considering this new E1E2 heterodimer structure [171]. 

## Figures and Tables

**Figure 1 viruses-15-01151-f001:**
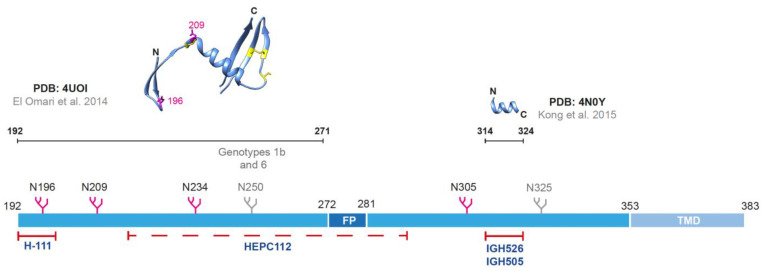
Structure of the HCV envelope glycoprotein E1. The E1 protein contains 190 aa (160 aa for the ectodomain and 30 aa for the transmembrane domain (TMD)), 5 or 6 N-glycosylation sites (including 4 which are highly conserved in all genotypes, represented in magenta; N250 exclusively found in genotypes 1b and 6, and N325 absent when a Proline residue is present immediately following the sequon (Asn-X-Ser/Thr), both represented in gray) and a putative fusion peptide (FP). The crystal structure of the N-terminal domain of E1 considered individually was determined (PDB: 4UOI), as well as the region 314–324 (PDB: 4N0Y) by co-crystallization with the human antibody IGH526. Antibodies (in dark blue) binding sites: aa 192–202 for the human monoclonal antibody (mAb) H-111, aa 215–299 for the human mAb HEPC112, and aa 313–324 for the human mAbs IGH505 and IGH526 [49,58]. Cysteines (and disulfide bonds) are represented in yellow and asparagines in magenta. Molecular graphics were performed using Chimera UCSF [59]. PDB: Protein Data Bank (https://www.rcsb.org/ accessed on 26 February 2023).

**Figure 3 viruses-15-01151-f003:**
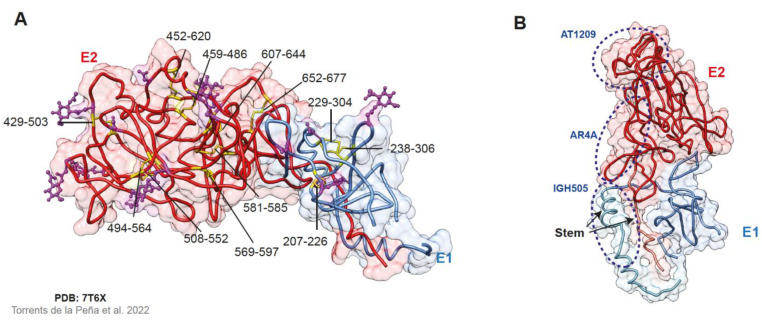
Structure of the E1E2 heterodimer. (**A**) Representation of the E1E2 complex (PDB: 7T6X), E1 in blue and E2 in red, with N-glycans in purple and disulfide bonds in yellow [51]. (**B**) In the “Stem-in-hand” model, the stem of E2 (aa 701–717) is held by E1, which represents the “hand”. The epitopes of antibodies used to solve the structure of the heterodimer are encircled in blue. AT1209 targets the CD81 binding loop in E2, AR4A is directed against the back layer of E2 but requires the presence of E1, and IGH505 binds amino acids in the stem region of E1. Molecular graphics were performed using Chimera UCSF [59]. PDB: Protein Data Bank (https://www.rcsb.org/ accessed on 26 February 2023).

**Table 1 viruses-15-01151-t001:** Summary of the residues and characteristics of the antigenic regions in the E2 protein, and the antigenic domains overlapping with these regions [114,115,121].

Antigenic Region	Contact Residues	Overlapping Domain	Characteristics
1	495, 519, 544, 545, 547, 548, 549 and 632	Some residues of domain C	Non-neutralizing region
2	597–645	A	Back layer and poor neutralizing region
3	396–424, 436–447, and 523–540	B, D, and E	Neutralizing region inducing bNAbs
4	201–206, 279, 487, 540, 547, 657, 658, 692, 698 *, 700	E1 antigenic site aa 192–207	Region comprising residues in E1 and E2 proteins, induction of bNAbs
5	201–206, 639 *, 657, 658, 665, 692	E1 antigenic site aa 192–207, and some residues of domain A	Region comprising residues in E1 and E2 proteins, induction of bNAbs

*: conserved residue.

## Data Availability

Not applicable.

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
