# Peer review of "Current Hepatitis C Vaccine Candidates Based on the Induction of Neutralizing Antibodies"

_viruses, 2023, doi:10.3390/v15051151_

Round 1
Reviewer 1 Report (Previous Reviewer 2)
The revised manuscript addresses my concerns. This is an excellent and timely review
acceptable
Author Response
Reviewer 1
The revised manuscript addresses my concerns. This is an excellent and timely review.
We would like to thank Reviewer 1 for this very positive view of our work. We are really pleased that our revised version of the manuscript has satisfactorily addressed the concerns of Reviewer 1.
Reviewer 2 Report (New Reviewer)
This manuscript by Gomez-Escobar and colleagues is a well-organized review of the biomedical literature with respect to structural determinants of hepatitis C virus (HCV) surface antigens as potential vaccine targets. These structural regions are composed of the major HCV spike proteins E1 and E2. The included tables are a strength of the manuscript. Moreover, epitopes that render neutralizing antibodies effective are rational targets for vaccine design based on a classical understanding of vaccinology. How to effectively elicit long-lasting and broadly neutralizing antibodies against HCV is another matter altogether. While the manuscript is a solid start, several issues should be addressed prior to publication.
1. The authors may wish to add additional rationale to support the continued need for an effective vaccine. Within the past year, a MMWR article published by the CDC shows that only about a third of patients with a HCV diagnosis and medical insurance receive DAA treatment (Source: https://www.cdc.gov/mmwr/volumes/71/wr/mm7132e1.htm?s_cid=mm7132e1_w).
2. It is clear that the scope and length of the manuscript do not allow for adequate coverage of the relevant immunologic principles and conundrums that have limited production of an efficacious HCV vaccine to date. That said, the issue is not simple, and the reference used as a justification for neutralizing antibodies (#35) is intended for a general scientific audience. Moreover, the citation is wholly inadequate in addressing the complicated issue of HCV vaccine immunology (e.g. immunodominance, mutation, longevity, prophylactic vs therapeutic approaches, etc.). Accordingly, the authors should direct readers to one of the many recent reviews on the topic and clarify that the issue is far more complicated than failure of a T-cell based vaccine.
3. Ideally, systematic reviews should contain documentation of the authors’ search strategy to demonstrate the comprehensiveness of the content and address possible missing components or other limitations. What were the criteria for eligibility and inclusion of the cited articles? Several vaccine candidates from clinical trials were mentioned. Does the manuscript address all current/proposed HCV vaccine trials listed on the ClinicalTrials.gov database?
Author Response
Reviewer 2
This manuscript by Gomez-Escobar and colleagues is a well-organized review of the biomedical literature with respect to structural determinants of hepatitis C virus (HCV) surface antigens as potential vaccine targets. These structural regions are composed of the major HCV spike proteins E1 and E2. The included tables are a strength of the manuscript. Moreover, epitopes that render neutralizing antibodies effective are rational targets for vaccine design based on a classical understanding of vaccinology. How to effectively elicit long-lasting and broadly neutralizing antibodies against HCV is another matter altogether. While the manuscript is a solid start, several issues should be addressed prior to publication.
We would like to thank Reviewer 2 for this constructive review of our work and his/her valuable comments to further improve the manuscript.
- The authors may wish to add additional rationale to support the continued need for an effective vaccine. Within the past year, a MMWR article published by the CDC shows that only about a third of patients with a HCV diagnosis and medical insurance receive DAA treatment (Source: https://www.cdc.gov/mmwr/volumes/71/wr/mm7132e1.htm?s_cid=mm7132e1_w).
Our response to this comment:
We thank reviewer 2 for this pertinent comment and this citation suggestion.
Although the current DAA-based treatments are very effective and control 95% of infections induced by the different HCV genotypes, they remain very expensive and are only distributed to a tiny part of the HCV-diagnosed population. As many countries continue to experience high transmission rates, there is an urgent need to develop an effective and affordable vaccine in order to control the global epidemic and reduce the burden on healthcare systems.
As suggested by reviewer 2, we now cite in the revised version of the manuscript (pages 1 and 2, lines 43-48) the MMWR article published by Thompson and collaborators in 2022 which illustrates this phenomenon and reports that in the USA only about one-third of patients with diagnosed HCV infection and medical insurance received DAA treatment and that treatment rates varied considerably by age and insurance payor:
“Current treatment based on direct-acting antivirals (DAAs) leads to elimination of HCV in more than 95% of the cases, but it has some limitations. The risk to develop HCC after treatment remains high, especially in patients with advanced liver fibrosis [1,17–21]. Resistance to DAAs was also observed in subjects infected with some rare HCV subtypes (1l, 4r, 3b, 3g, 6u, 6v) that emerged in specific geographical zones [17,22,23]. DAAs do not protect from reinfection, and so intravenous drug users (IVDUs), which are frequently exposed to HCV, remain at high risk of reinfection after DAA-mediated cure [24–26]. Beyond that, DAA-based treatments remain expensive for patients if not covered by a national program or by a personal medical insurance. A recent study in the USA showed that treatment rates varied considerably by age and insurance payor with the result that only one-third patients with an HCV diagnosis and medical insurance receive DAA treatment [27]. Thus, generation of a vaccine against this virus will help to control its transmission, such as in high-risk populations, and to respond to the current limitations of treatment [28–30].”
- It is clear that the scope and length of the manuscript do not allow for adequate coverage of the relevant immunologic principles and conundrums that have limited production of an efficacious HCV vaccine to date. That said, the issue is not simple, and the reference used as a justification for neutralizing antibodies (#35) is intended for a general scientific audience. Moreover, the citation is wholly inadequate in addressing the complicated issue of HCV vaccine immunology (e.g. immunodominance, mutation, longevity, prophylactic vs therapeutic approaches, etc.). Accordingly, the authors should direct readers to one of the many recent reviews on the topic and clarify that the issue is far more complicated than failure of a T-cell based vaccine.
Our response to this comment:
We agree with this important comment from Reviewer 2.
We recognize that the reference used as a justification for neutralizing antibodies is not the most appropriate to address the complex issue of HCV vaccine immunology and that we were somewhat clumsy in discussing the reasons for the failure of the T-cell based vaccine. This failure is obviously not solely explained by the absence of induction of broadly neutralizing antibodies, and the issue is much more complex.
In the revised version of this manuscript, we have modified the text to clarify these different points and we have mainly relied on the work of Phelps and collaborators, published in 2021 in the journal Viruses. In this study, they examined all the published data from this first efficacy trial as well as those from previous clinical and pre-clinical studies of the vaccine candidate, and highlighted the different key elements that become important for the development of an HCV vaccine, such as the genetic diversity, the induction of humoral responses characterized by broadly neutralizing antibodies and the induction of cytotoxic and helper T cell responses.
- Page 2, lines 62-73:
“To date, the most advanced HCV vaccine candidate is a T-cell vaccine consisting on a prime-boost regimen with two different viral vectors that encode the genotype (Gt) 1b (BK strain) HCV non-structural proteins NS3-5B (mutated in the NS5B gene to abolish the RNA polymerase activity) (NSmut) [31]. This vaccine was first shown to induce strong cellular responses in chimpanzees and protect 80% of them from challenge with HCV [32]. Then, safety and efficacy of various viral vectors encoding the NSmut construct (adenovirus 6, chimpanzee adenovirus 3 (ChAd3) and modified vaccinia Ankara (MVA)) were evaluated in healthy volunteers (clinical trials: NCT01070407 and NCT01296451) [33,34]. Barnes and collaborators found that the vaccine was well tolerated with no severe adverse effects and led to the generation of cellular responses, especially when using the MVA-NSmut, as booster, which induced strong and sustained CD4+ T cell responses overtime [33,34]. However, in the latest randomized clinical trial phase 1/2 (NCT01436357), in which 274 participants (IVDUs) followed the prime-boost regimen ChAd3-NSmut / MVA-NSmut, vaccination did not prevent the development of chronic infection [35]. These results suggest that humoral responses characterized by broadly neutralizing antibodies (bNAbs), along with cytotoxic and helper T cell responses, as well as the conception of novel immunogens that generate immune responses against genetically diverse HCV genotypes/subtypes should be considered in HCV vaccine development, as discussed in a recent review [36]. Our review focuses on the structural components needed for the induction of neutralizing antibodies (NAbs) and the current status of HCV envelope-based vaccine candidates aiming to elicit humoral responses.
2. The envelope glycoproteins as target of neutralizing antibodies
The HCV envelope glycoproteins (E1 and E2) constitute the main targets of NAbs. These proteins are highly glycosylated (5-6 and 11 N-glycans, respectively) transmembrane proteins type I, anchored to the endoplasmic reticulum (ER)-derived membrane by a 30-amino acid (aa) transmembrane domain (TMD), and located at the surface of HCV [38].”
- Ideally, systematic reviews should contain documentation of the authors’ search strategy to demonstrate the comprehensiveness of the content and address possible missing components or other limitations. What were the criteria for eligibility and inclusion of the cited articles? Several vaccine candidates from clinical trials were mentioned. Does the manuscript address all current/proposed HCV vaccine trials listed on the ClinicalTrials.gov database?
Our response to this comment:
We did not attempt to describe all vaccine candidates actually in clinical trials but rather to list all vaccine candidates based on HCV envelope proteins, and therefore likely to induce neutralizing antibodies, whether in development or in some cases in clinical trials.
To clarify this point, we have modified the sentence, page 2 lines 69-71:
“Our review focuses on the structural components needed for the induction of neutralizing antibodies (NAbs) and the current status of HCV envelope-based vaccine candidates aiming to elicit humoral responses.”
Round 2
Reviewer 2 Report (New Reviewer)
The authors have adequately addressed all of my major concerns. I defer to the editors on whether or not the search strategy, inclusion criteria, and potential limitations should be disclosed in the review.
This manuscript is a resubmission of an earlier submission. The following is a list of the peer review reports and author responses from that submission.
Round 1
Reviewer 1 Report
This review article by Gomez-Escobar nicely summarizes the current knowledge on the structure of the HCV E1 and E2 envelope proteins and the link between certain regions within the envelope proteins and monoclonal antibodies with or without neutralizing activity; and antibodies generated during natural infection. The second part of this review focuses on the different strategies employed for the development of a B-cell vaccine.
Overall this review provides a comprehensive overview with up to date information. It is nicely structured and reads fluently. I am confident that the content of this manuscript is timely and will probably be highly cited in the near future.
I only have the following minor comments:
1) Two monoclonal antibodies that were more recently isolated from chronically infected individuals are missing in the overview. The first antibody, designated A6, targets an epitope in E1 spanning AA230-239 and is non-neutralizing (Mesalam et al., Virology 2018). A second antibody, designated 2A5, targets a conformational epitope with its main epitope spanning AA434-446 (epitope II) but also additional regions/contact residues are involved (Desombere et al., Antiviral Research 2017). The latter antibody was shown to have neutralizing capacity both in vitro and in vivo.
2) Likewise two older antibodies targeting epitope III were isolated from an infected individual, specifically 1:7 and A8 (Johansson et al., PNAS 2007).
3) Finally, in the section regarding HVR1 the authors correctly state that deletion of HVR1 renders the virus more susceptible to neutralizing antibodies. The work of Bankwitz and colleagues should be included here (Bankwitz et al., JVI 2010).
For completeness of this review I would propose to include the above mentioned work.
Author Response
Reviewer 1
This review article by Gomez-Escobar nicely summarizes the current knowledge on the structure of the HCV E1 and E2 envelope proteins and the link between certain regions within the envelope proteins and monoclonal antibodies with or without neutralizing activity; and antibodies generated during natural infection. The second part of this review focuses on the different strategies employed for the development of a B-cell vaccine.
Overall this review provides a comprehensive overview with up to date information. It is nicely structured and reads fluently. I am confident that the content of this manuscript is timely and will probably be highly cited in the near future.
We would like to thank Reviewer 1 for this very positive view of our work.
I only have the following minor comments:
1) Two monoclonal antibodies that were more recently isolated from chronically infected individuals are missing in the overview. The first antibody, designated A6, targets an epitope in E1 spanning AA230-239 and is non-neutralizing (Mesalam et al., Virology 2018). A second antibody, designated 2A5, targets a conformational epitope with its main epitope spanning AA434-446 (epitope II) but also additional regions/contact residues are involved (Desombere et al., Antiviral Research 2017). The latter antibody was shown to have neutralizing capacity both in vitro and in vivo.
Our response to this comment:
As requested, we mentioned the mAbs A6 and 2A5 in the revised version of the manuscript in the corresponding sections, AS112 antigenic site of E1 (page 4 lines 138-141) and the epitope II of E2 (page 6 lines 239-243), respectively:
“The mAb A6, characterized by Mesalam and colleagues, also targets a linear epitope within the AS112 (aa 230-239). However, this antibody isolated from Gt 1b HCV-infected patient does not exhibit neutralizing activity [63].”
“Another antibody targeting this site is the mAb 2A5 (conformational epitope within the aa 434-446 region) which was isolated from a patient chronically infected with Gt 1b HCV. This antibody neutralizes several genotypes and strains of HCV in the HCVpp and HCVcc systems, as well as protects humanized mice against HCV challenge [106].”
As our review is specifically focused on antibodies capable of neutralizing HCV, we have however decided to represent in Figure 2 only the 2A5 antibody which exhibits neutralizing properties.
2) Likewise two older antibodies targeting epitope III were isolated from an infected individual, specifically 1:7 and A8 (Johansson et al., PNAS 2007).
Our response to this comment:
We agree that these antibodies are of great importance because they block the binding of E2 protein to the CD81 receptor, and they possess broad-neutralizing properties. This information was added page 6 lines 245-253 and included in Figure 2:
“Because of the importance of this site in the viral entry step through the CD81 receptor, blocking the E2-CD81 interaction is a main target of bNAbs. In the study from Torrents de la Peña and collaborators, the bNAb AT1209, which targets a region comprising the CD81bl and the front layer, was used to determine the structure of E1E2 [107]. They revealed that the aa critical for the interaction between E2 and CD81 overlap with those located in the epitope of AT1209 [50]. Among the bNAbs targeting the CD81 are those reported by Johansson and colleagues, mAbs 1:7 and A8, which were isolated from a Gt 2b HCV-infected subject and able to recognize an epitope in the region spanning residues aa 523–535 [108].”
“Figure 2. Structure of the HCV envelope glycoprotein E2. E2 is a 360 aa protein (including 30 aa for the transmembrane domain (TMD)) that contains 3 variable regions (hypervariable region (HVR)1, HVR2 and intergenotypic variable region (igVR)), a front layer, a back layer, a CD81-binding loop (CD81bl), a stem region and 11 N-glycosylation sites (in magenta). The crystal structure of the core of E2 considered individually (aa 421-645/656) was obtained by two groups (PDB: 4MWF and 4WEB). Three epitopes for neutralizing antibodies were identified (I, II and III). The epitope I (aa 412-423) can adopt 3 conformations: β-hairpin (PDB: 4DGY), semi-open (PDB: 4XVJ) and open (PDB: 4WHY). The epitope II (aa 434-446) was co-crystallized with the human monoclonal antibodies (mAbs) HC.84-27 (PDB: 4JZO) and HC.84-1 (PDB: 4JZN), but also targeted by the 2A5 mAb. The crystal structure of the epitope III (aa 523-535), inside the CD81bl, (PDB: 5NPJ) was also obtained by co-crystallization with the mouse mAb DAO5, and targeted by the mAbs 1:7 and A8. Antibodies targeting each epitope are indicated in dark blue. Cysteines (and disulfide bonds) are represented in yellow and asparagines in magenta. Molecular graphics were performed using Chimera UCSF [57]. PDB: Protein Data Bank (https://www.rcsb.org/)”
3) Finally, in the section regarding HVR1 the authors correctly state that deletion of HVR1 renders the virus more susceptible to neutralizing antibodies. The work of Bankwitz and colleagues should be included here (Bankwitz et al., JVI 2010). For completeness of this review I would propose to include the above mentioned work.
Our response to this comment:
We agree with this observation. We included the work from Bankwitz and colleagues, as well as an example of how the deletion of HVR1 does not necessarily translate in an improvement of immunogenicity in the vaccine development context. These changes are included in the revised version of the manuscript page 4 lines 182-187:
“Interestingly, the deletion of the HVR1 was reported to increase the susceptibility to neutralization by NAbs of virions, suggesting a shielding effect of antigenic sites in the E2 protein by the HVR1 [81,82]. In a recent study of a vaccine candidate based on inactivated recombinant HCVcc, deletion of the HVR1 led to increased accessibility of NAbs (AR3A and AR4A), but did not result in increased immunogenicity suggesting a much complex role of this region [83,84].”
Reviewer 2 Report
There is one major criticism. In Section 2 (the envelope glycoproteins as target of neutralizing antibodies) does not take into account the recent structural advances on E2/CD81 and E1E2. As a result there are several inconsistencies in those sections. For example, the lack of a structure for full length E1 and the function of CD81 binding loop (epitope III) during entry. In fact, the last sentence of the manuscript states that the E1E2 structure will assist in vaccine design without any further explanation of how or why. The review should discuss these recent findings. Do these results have implications in vaccine design? If so, how?
Author Response
Reviewer 2
There is one major criticism. In Section 2 (the envelope glycoproteins as target of neutralizing antibodies) does not take into account the recent structural advances on E2/CD81 and E1E2. As a result there are several inconsistencies in those sections. For example, the lack of a structure for full length E1 and the function of CD81 binding loop (epitope III) during entry. In fact, the last sentence of the manuscript states that the E1E2 structure will assist in vaccine design without any further explanation of how or why. The review should discuss these recent findings. Do these results have implications in vaccine design? If so, how?
We would like to thank Reviewer 2 for this critical and constructive review of our work and his/her valuable comments to further improve the manuscript.
We agree that very recent structural findings in our review were missing, particularly relating to the E2/CD81 interaction and the structure of the full-length E1E2 heterodimer, these works having been published just at the time or just after the time we submitted our manuscript. Therefore, in the revised version of our manuscript we included several passages containing the main outcomes of those works.
First, the findings of Kumar and colleagues regarding the structural characterization of the E2-CD81 interaction were included in the subsection 2.2. The envelope glycoprotein E2, page 4 lines 146-151:
“The interaction between E2 and the CD81 receptor was recently characterized by Kumar and colleagues who proposed a docking model in which the residues 418-422 in E2 are displaced and allow the extension of an internal loop spanning the residues 520-539, which approaches Tyr529 and Tyr531 to the membrane in preparation for a low-pH-mediated fusion [67]. In a complementary study, Kumar and colleagues reported that for proper interaction with CD81, the front layer and the AS412 in E2 are essential [68].”
Then, the new structural elements reported by Torrents de la Peña and collaborators were added in the different sections of the review:
1. A new subsection (3. The E1E2 heterodimer) was created to describe some studies aiming to characterize the E1E2 heterodimer using different strategies. This can be found in pages 7 and 8 lines 286-316:
“2.3. The E1E2 heterodimer
The structural characterization of the E1E2 heterodimer has been hindered by the challenging purification process of these transmembrane proteins. So, the structure of truncated versions of E1E2 and in silico models were proposed. That is the case of Cao and collaborators that designed a functional heterodimer with E1E2 ectodomains (Gt 1b, strain Con1) fused with a Fc-tag; this complex could be recognized by NAbs such as AR3A, HCV-1 and IGH526, and bound some cellular receptors [120]. The in silico models predicted the E1E2 complex structure using computational tools such as Rosetta [55,56]. Other studies aimed to determine the role of every residue in the HCV envelope glycoproteins following mutagenesis strategies [111,121]. For instance, Gopal and colleagues found out that residues in the back layer of E2, within the heterodimer, can impact the binding to CD81 and NAbs [111]. A similar comprehensive mutagenesis study reported that 57% (311/545) of the mutants resulted in non-functional HCV envelope glycoproteins and 92% of the mutations led to a change in viral functionality, suggesting that HCV high rate of replication must be a mechanism to compensate the vulnerability of its envelope glycoproteins [121].
Lately, Torrents de la Peña and collaborators determined the structure of the full-length E1E2 heterodimer Gt 1a (AMS0232 strain) in complex with Fabs from AR4A, AT1209 and IGH505 NAbs (Figure 3) [50]. In this study, the authors were able to model 51% of E1 and 82% of E2 comprising the E1/E2 interface, the disulfide networks of E1 and E2, the E1/E2 glycan shield and the epitopes of the NAbs used, while the unresolved TMDs were predicted using AlphaFold (Figure 3A). Torrents de la Peña and colleagues proposed a “stem-in-hand” model in which the ectodomain of the E1 protein wraps the E2 stem and interacts with the base of E2 (Figure 3B). They reported that the non-covalent interactions (hydrophobic and hydrogen bonds) observed between E1 and E2 were consistent with previous studies [122,123] and that the glycans N196 and N305 reinforced this interaction. Despite the prior identification of highly conserved cysteines residues, the disulfide patterns were different from previously determined structures. Thus, the authors hypothesized that the proximity of cysteines in the HCV envelope glycoproteins may allow disulfide bond scrambling [50]. These findings will contribute to the design of immunogens possessing the best structural characteristics to induce bNAbs.”
2. The updated structural information related specifically to the E1 protein was added in the subsection 1.focused on this envelope glycoprotein at different places:
- Page 2 lines 86-91:
“The crystal structure of the full-length E1 protein was unknown until not long ago because the expression of E1 in the absence of the E2 protein can lead to protein aggregation [45–47]. Thus, El Omari and colleagues solved the N-terminal domain of a Gt 1 (H77 strain) E1 protein through crystallization in low-pH conditions, representing a post-attachment conformation of the domain (Figure 1) [48].”
- Pages 2 and 3 lines 95-106:
“However, in the latest study by Torrents de la Peña and collaborators [50], the conformation of the N-terminal domain of E1 protein, within the heterodimer, differed from the structure determined by El Omari and colleagues [48]. This finding may confirm that E1 requires E2 protein for correct folding. The study also confirmed the presence of 4 disulfide bonds and the potential N-glycosylation sites with the exception of N325 (Figure 1), previously described to be absent when a proline residue is present immediately following the sequon (Asn-X-Ser/Thr) [51]. The E1 protein has as well the ability to form trimeric structures through the GxxxG motif located within its TMD [52], the same motif that was suggested to be involved in the interaction between the TMDs of both envelope proteins [53,54]. This trimeric arrangement of the E1E2 heterodimers has been confirmed by using computational and biological models [55,56] as well as suggested by the lack of glycans in a hydrophobic region on the structure of the recently solved E1E2 complex [50].”
- Pages 3 and 4 lines 131-135:
“While the epitope of the mAb IGH505 was defined in complex with the E1E2 heterodimer and found to target the surface-exposed conserved a-helix in E1 (H316, W320, M323, M324) [50]. Because of the location of the epitopes of both antibodies (IGH526 and IGH505) in E1, it was proposed that neutralization may occur by impeding conformational changes in the heterodimer [50].”
The N325 glycan was also added to the Figure 1, even though it can be absent as Torrents de la Peña and collaborators observed on their model.
“Figure 1. Structure of the HCV envelope glycoprotein E1. The E1 protein contains 190 aa (160 aa for the ectodomain and 30 aa for the transmembrane domain (TMD)), 5 or 6 N-glycosylation sites (including 4 which are highly conserved in all genotypes, represented in magenta; N250 exclusively found in genotypes 1b and 6, and N325 absent when a Proline residue is present immediately following the sequon (Asn-X-Ser/Thr), both represented in gray) and a putative fusion peptide (FP). The crystal structure of the N-terminal domain of E1 considered individually was determined (PDB: 4UOI), as well as the region 314-324 (PDB: 4N0Y) by co-crystallization with the human antibody IGH526. Antibodies (in dark blue) binding sites: aa 192-202 for the human monoclonal antibody (mAb) H-111, aa 215-299 for the human mAb HEPC112, and aa 313-324 for the human mAbs IGH505 and IGH526. Cysteines (and disulfide bonds) are represented in yellow and asparagines in magenta. Molecular graphics were performed using Chimera UCSF [57]. PDB: Protein Data Bank (https://www.rcsb.org/)”
3. The updated structural characterization of the E2 protein was included in the subsection 2.2. dedicated to the description of this envelope protein:
- Page 4 lines 164-173:
“The structure of the full-length E2 protein was recently determined within the E1E2 heterodimer Gt 1a by Torrents de la Peña and collaborators [50]. This study resolved the structure of 2 new regions in the E2 protein: the base (aa 645-700) and the stem (aa 701-717), which connects base with TMD. It also confirmed that E2 has 9 disulfide bonds, consistent with previous studies [71], but differed from the structure obtained in complex by Kong and collaborators due to disulfide-bond scrambling [74]. As reported for E1, they observed as well all the potential N-glycosylation sites in E2. However, a noncanonical NXV motif at the N695 glycosylation site was reported with and without binding to AR4A and AT1209 Fabs, and its removal led to a slight increase of the viral infectivity using HCVpp [50].”
- Page 6 lines 245-253:
“Because of the importance of this site in the viral entry step through the CD81 receptor, blocking the E2-CD81 interaction is a main target of bNAbs. In the study from Torrents de la Peña and collaborators, the bNAb AT1209, which targets a region comprising the CD81bl and the front layer, was used to determine the structure of E1E2 [107]. They revealed that the aa critical for the interaction between E2 and CD81 overlap with those located in the epitope of AT1209 [50]. Among the bNAbs targeting the CD81 are those reported by Johansson and colleagues, mAbs 1:7 and A8, which were isolated from a Gt 2b HCV-infected subject and able to recognize an epitope in the region spanning residues aa 523–535 [108].”
- Page 7 lines 269-273:
“The AR4 contains as well epitopes of bNAbs like AR4A, which was used in the characterization of the full-length E1E2 heterodimer. Torrents de la Peña and colleagues reported that the E1E2 complex bound to AR4A was stabilized in a prefusion conformation suggesting that neutralization with this antibody occurs by hindering conformational changes needed for fusion [50].”
Furthermore, we included a recent publication proposing a vaccine candidate based on the structural findings of Torrents de la Peña and collaborators. In the design of the immunogens, Sliepen and colleagues took advantage of the knowledge regarding the positioning of the HCV envelope proteins and proposed permutated E2E1 nanoparticles that can induce slightly increased cross-neutralizing antibodies than the E2 protein alone. This is a clear example of how understanding the structure of proteins may facilitate the rationally designed protein-based vaccines. This information was included in the subsection 4.1. Vaccines using both HCV envelope glycoproteins (E1 and E2), page 11 lines 468-480:
“The nanoparticles are commonly used as scaffolds for the display and delivery of immunogens to antigen-presenting cells (reviewed in [167]). In their study, Sliepen and collaborators proposed trimeric permutated E2E1 Gt 1a (strain AMS0232) nanoparticles [168]. HCV envelope glycoproteins were permutated to bring closer the N-terminus of E1 and C-terminus of E2 in order to facilitate interactions between E1 and E2 proteins [50]. E2E1 nanoparticles were used to immunize New Zealand female rabbits, which developed antibodies able to neutralize more HCVpp than a monomeric E2 protein. E2E1 mosaic nanoparticles were also efficiently generated (harboring E2E1 from different genotypes: strains H77, AMS2b, AMS3a, UKNP4.1.1, UKNP5.2.1, UKNP6.1.2) with the objective of focusing the responses to epitopes that are conserved among the antigens presented. After immunization, the mosaic nanoparticles induced slightly better cross-neutralizing antibodies (6 HCVpp) than a cocktail of 6 E2E1 nanoparticle harboring each of them proteins from one strain [168].”
This E2E1 nanoparticle vaccine candidate was included in the Table 2 that summarizes the described B-cell vaccines against HCV in this review.
In the last sentence of our conclusion, we explained the implication of these structural studies in vaccine development. First, as we observe in Table 2, most of the groups working on the development of a vaccine have characterized the elicited NAbs only as anti-E1/anti-E2, but not all of them against to specific antigenic domains and regions in the E1E2 heterodimer. Therefore, the recent structural studies, describing numerous epitopes and antigenic regions/domains, may enable an improved characterization of the antibodies induced against the regions highlighted by these studies. Finally, we added the study of Sliepen and colleagues as an example of the interest of resolving the structure of envelope proteins in the vaccine development.
- Page 16 lines 608-614:
“Furthermore, findings from structural studies may contribute to improve the definition of NAbs epitopes induced by vaccination, which are frequently merely characterized as anti-E1/anti-E2 (Table 2). Finally, the recent publication of the E1E2 heterodimer structure will enable the design of better immunogens containing the structural elements needed for E1E2 complex stabilization and epitopes presentation [50], such as the E2E1 nanoparticles-based vaccine candidate that was designed considering this new E1E2 heterodimer structure [168].”